# Self-Diagnostic Opportunities for Battery Systems in Electric and Hybrid Vehicles

**Szabolcs Kocsis Szürke [1], Gergő Sütheö [2], Péter Őri [1] and István Lakatos [1,\***

[1]  Central Campus Győr, Széchenyi István University, H-9026 Győr, Hungary
[2]  Zalaegerszeg Innovation Park, Széchenyi István University, H-8900 Zalaegerszeg, Hungary; sutheo.gergo@ga.sze.hu
[\*]  Correspondence: lakatos@sze.hu

**Abstract:** The number of battery systems is also growing significantly along with the rise in electric and hybrid car sales. Different vehicles use different types and numbers of batteries. Furthermore, the layout and operation of the control and protection electronics units may also differ. The research aims to develop an approach that can autonomously detect and localize the weakest cells. The method was validated by testing the battery systems of three different VW e-Golf electric vehicles. A wide-range discharge test was performed to examine the condition assessment and select the appropriate state of charge (SoC) for all three vehicles. On the one hand, the analysis investigated the cell voltage deviations from the average; the tests cover deviations of 0 mV, 12 mV, 60 mV, 120 mV, and 240 mV. On the other hand, the mean value calculation was used to filter out possible erroneous values. Another important aspect was examining the relationship between the state of charges (SoC) and the deviations. Therefore, the 10% step changes were tested to see which SoC level exhibited more significant voltage deviations. Based on the results, it was observed that there are differences between the cases, and the critical range is not necessarily at the lowest SoC level. Furthermore, the load rate (current) and time of its occurrence play an important role in the search for a faulty cell. An additional advantage of this approach is that the process currently being tested on the VW e-Golf can be relatively simply transferred to other types of vehicles. It can also be a very useful addition for autonomous vehicles, as it can self-test the cells in the system at low power consumption.

**Keywords:** battery system; SoC; dynamic testing; fault location algorithm; autonomous fault detection





## 1. Introduction

Recently, there has been an apparent increase in the use of electric vehicles in land, air, and water transport. This is mainly driven by global efforts to reduce carbon emissions and mitigate the effects of climate change. Government initiatives such as green treaties and sustainability legislation play a key role in this process [1]. This growth is partly due to advanced battery technology, the expansion of charging infrastructure, and increased consumer awareness of electric vehicles [2,3]. The significant change in air transport due to the increased use of Unmanned Aerial Vehicles (UAVs) covers a wide range of applications, including commercial photography, agricultural surveys, and disaster relief [4–6]. The proliferation of electric propulsion in waterborne transport, particularly for inland waterways and short-sea shipping, has been identified as a sustainable alternative to conventional systems. Introducing such systems can significantly reduce the environmental impact while increasing the efficiency of maritime transport operations [7–9]. Battery systems play a crucial role in the energy storage of electric vehicles (EVs), whether used for land, air, or water transport. The differences in battery systems between these transport modes are significant and tailored to each vehicle type's specific requirements and applications [10]. The automotive industry continuously develops lithium-ion batteries to increase range, reduce charging time, and improve battery safety [11]. Electric aircraft also use lithium-based

batteries, with even greater emphasis on energy density and weight reduction [12]. Electric watercraft often use larger, more massive battery systems that can provide power for more extended periods of time [13].

The energetics of autonomous vehicles may differ from conventional vehicles in many ways, as their design and operation present new challenges in terms of energy efficiency and energy management. The integration of various electronic systems, such as edge computing, artificial intelligence (AI), and advanced driver assistance systems (ADAS), significantly affects the power consumption and overall energy efficiency of autonomous vehicles [14,15]. In addition, deploying numerous sensors and computing resources on board autonomous vehicles significantly increases the continuous vehicle load, resulting in higher power consumption [16]. Furthermore, using deep learning approaches in autonomous vehicles introduces high computational complexity and power consumption, which may affect the driving range of these vehicles [15,17]. In addition, integrating electronic components in autonomous vehicles has implications for fuel consumption and environmental sustainability. Studies have shown that autonomous vehicles have the potential to reduce fuel consumption by reducing traffic congestion and improving road safety, thereby contributing to energy efficiency and environmental protection [18,19]. From the diagnostic point of view of autonomous vehicles, the integrated self-diagnostic systems (ISS) built for this purpose are based on the Internet of Things (IoT). The system collects information from the autonomous vehicle's sensors and uses deep learning to inform the driver of relevant results, thereby increasing operational safety [20].

Electric autonomous vehicles require complex diagnostics and self-diagnostics to ensure battery system safety and longevity. Implementing a real-time fault diagnosis and protection system is crucial for operational safety and performance. Fault diagnosis is often performed by monitoring the SoC to provide the necessary protection and self-healing operations [21]. Different methods have been analyzed for estimating SoC, technical state, and operational state by various authors and researchers [22]. Accurate state estimation and reliable prediction of remaining useful life (RUL) are critical issues [23–25]. Therefore, new data-driven techniques, including machine learning algorithms, are being applied in vehicle onboard diagnostic systems [26]. Developing and searching for diagnostic methods for SoH indicators is an important research area for optimizing battery management and energy efficiency [27]. To ensure smooth operation and monitoring of processes, an intelligent battery management system (BMS) is required to ensure that the battery has sufficient energy even in critical situations [28]. The BMS is essential for the safe and reliable operation of batteries in electric and hybrid electric vehicles. The BMS monitors the battery's state, including its SoC, health, and life, and controls charging and cell balancing. The performance of a battery can vary significantly under different operating and environmental conditions, which presents challenges for the implementation of the BMS [29].

During vehicle operation, some cells age faster than others, compromising the battery pack's capacity and lifetime [30]. One way to investigate cell degradation is to monitor the increase in internal resistance. This can be carried out using a model that includes parameters that depend on the degradation [31]. In the model, the degradation index can be determined from the value of the internal resistance corrected (e.g., for battery temperature) [32–34]. In addition, research is also being conducted to compare OBD methods with measurements taken directly on batteries, which can provide further insight into the efficiency and accuracy of diagnostic processes [35]. Specifically, based on communication via the OBD port, diagnostic analyses on an e-Golf vehicle were performed to identify and localize weak battery cells. This approach combines degradation modeling and real-time diagnostics [36].

Several fault diagnosis techniques have been proposed, including real-time voltage analysis [37], entropy-based approaches [38], and statistical distribution models [39]. These methods use advanced technologies such as artificial intelligence, statistical analysis, and entropy calculations to identify anomalies and faults in battery systems. In addition, fault

detection methods based on big data analysis have been explored to improve the accuracy of battery fault diagnosis in EVs [40].

Some battery diagnostic methods use a cyber-physical approach to detect vehicle anomalies with an efficiency of around 86%, but increasing cyber components also introduce new risks, leading to recalls [41]. Early prediction of battery thermal catastrophe and ignition can also be made using data-driven methods that characterize internal parameters and detect faults by analyzing each cell's real-time state. These methods can accurately identify faulty cells, detect problems early, and provide early warning of the risk of thermal disaster [32,42].

When electric vehicles are used, vibrations can cause insulation failures, which can endanger the safety of drivers and passengers. Therefore, it is important to check the real-time condition of the insulation between high voltage and ground for safe operation [43].

Monitoring and diagnosing battery conditions are essential to maximizing optimal performance and uptime. This becomes even more important for autonomous vehicles, as the vehicle needs to be able to make autonomous decisions that directly affect safety and efficiency. In our research, identifying faulty cells is a crucial aspect of battery diagnostics.

The tests aimed to use the faulty (weaker) cell search algorithm over a wide range of SoCs. The research aims not to implement or create a new battery model but rather to develop a diagnostic method that will allow the weak cells in the system to be located without having to disassemble the vehicle, thus avoiding the loss of warranty. The approach is currently being tested on electric vehicles, aiming to develop a method that can be applied to autonomous vehicles, that is as easy as possible to adopt, and that can even be operated as a stand-alone unit. The research focuses on diagnosing energy efficiency and battery conditions without directly affecting the vehicle's control system. The method is intended to reduce the computational requirements, which could positively impact energy management and, thus, the range of vehicles.

The initial discharge level ranged from 95% to 84% in the tests, and the lower discharge threshold was 10%. The tests were analyzed in five tests on three different vehicles.

On the one hand, deviations in the cell voltages from the average were analyzed, with tests covering deviations of 0 mV, 12 mV, 60 mV, 120 mV, and 240 mV. Another important aspect was the analyses as a function of charge level, where the perimeter was analyzed to see how the deviations were related to the SoC.

Performing continuous cell-level diagnostics can require large computing power. However, an autonomous vehicle can run self-diagnostics at specific intervals to assess the cells' status. To run the process faster, running the test at the SoC level makes sense, where there is a higher probability of significant cell voltage deviations. In our paper, we focus on defining this range.

However, it is important to note that the tests have been carried out on battery systems (vehicles) in good condition and that faulty systems (cells) are not included in the analyses. Currently, research is not investigating large cell voltage deviations; instead, the focus is on detecting smaller deviations. In the case of larger deviations, the operation of the vehicle's internal diagnostics is currently the guiding principle.

The second section describes the main parameters to be measured and the test procedure. The third section presents the evaluation of the results and the results obtained from the different measurements. The fourth section presents the conclusions and applicability of the results and their limitations.

## 2. Materials and Methods

The Volkswagen e-Golf second-generation battery system used for the tests has a total capacity of 35.8 kWh. The Samsung SDI-equipped lithium–nickel–manganese–cobalt (NMC) cells have a nominal voltage of 3.665 V. The e-Golf has two different modules, consisting of 6 and 12 cells. Cells are connected in triples in parallel, and such units are connected in series. The smaller module has a nominal voltage of $2 \times 3.665 = 7.33$ V, and the larger module has a nominal voltage of $4 \times 3.665 = 14.67$ V. The ten smaller modules

are in the middle of the battery pack, and the 17 large modules are at the edges. The total battery pack, therefore, has a total voltage of 325 V (88s3p configuration) when the 88 triple blocks are connected in series. The following diagram (Figure 1) shows the battery system of the e-Golf:

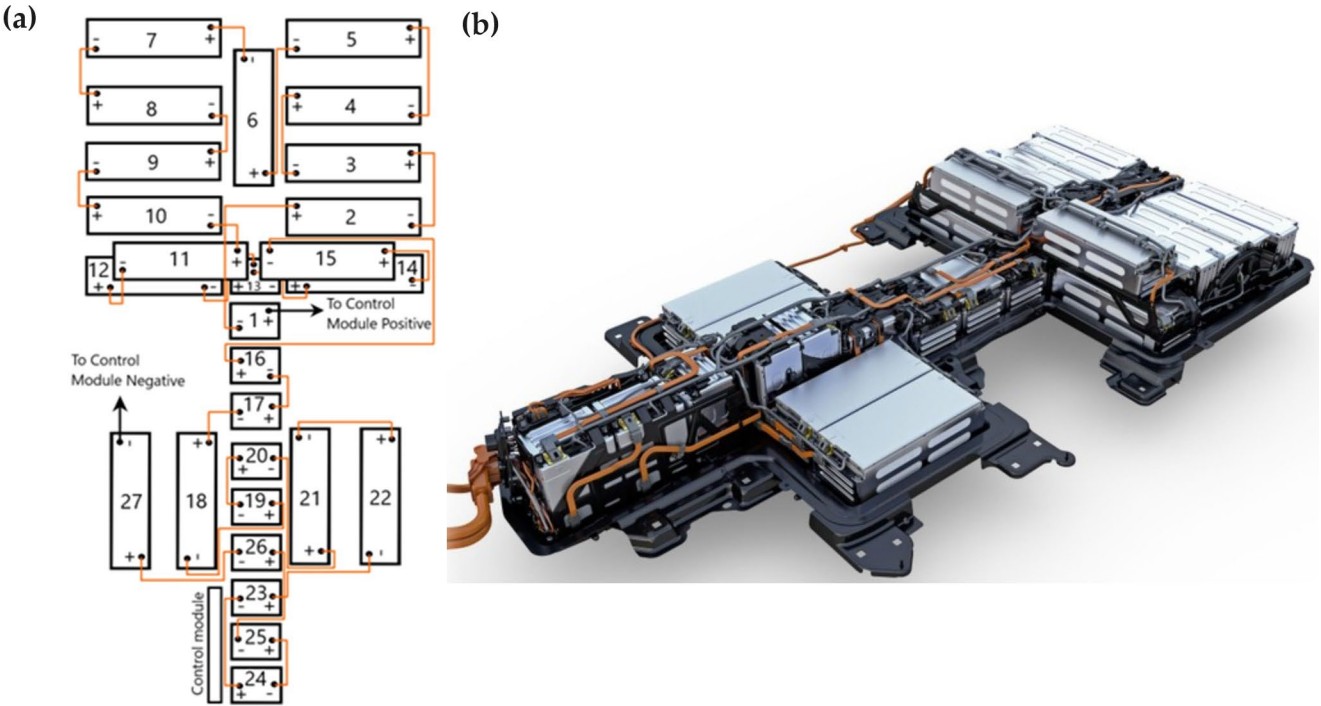

**Figure 1.** Volkswagen e-Golf battery system. Figure 1 (**a**) shows the schematic diagram of the Volkswagen e-Golf battery pack (**b**) shows the vehicle's battery system.

### 2.1. Measuring Instruments and Data Extraction Method

During the diagnostic measurements, it is possible to save 120 different signals related to the e-Golf's high-voltage battery from the vehicle's CAN communication network, which is: system voltage [V]; system current [A]; 88 cell voltages [V]; 27 module temperatures [°C]; charge level [%]; vehicle speed [km/h]; GPS speed [km/h].

Data will be collected from the CAN communication network using a dedicated CAN protocol-based measurement system based on Kvaser Memorator R SemiPro CAN USB interface (Mölndal, Sweden); AEM 30-2206 VDM GPS and accelerometer sensor (Hawtrone, CA, USA); 120 Ω resistors (Palmdale, CA, USA); ELM-327 OBDII connector socket (London, ON, Canada).

Figure 2 shows the assembled CAN protocol-based measurement system. It can be sampled at the standard bitrate available on the vehicle communication network at a bus speed of 500 kbit/s.

A serial query structure is used for data collection, meaning a response code is received from the ECU via a request code. It is important to consider the time elapsed between messages, as the ECU may ignore a query that arrives too frequently, causing data loss. Based on preliminary tests, 0.01 s is sufficient, but for safety reasons, it is better to leave 0.012 s between each message so that the maximum achievable sampling rate is 100 Hz. The process is shown in the flowchart in Figure 3.

According to Figure 3, the raw data are sorted and then grouped with the corresponding electric vehicle. This is followed by filtering and cleaning processes using a decoding file to process the CAN messages. After processing the data, the following values are available: system and cell voltage, current, temperature, and speed. Cell voltage deviations are determined in the evaluation phase. In the last step, the deviations are assigned to the SoC level.

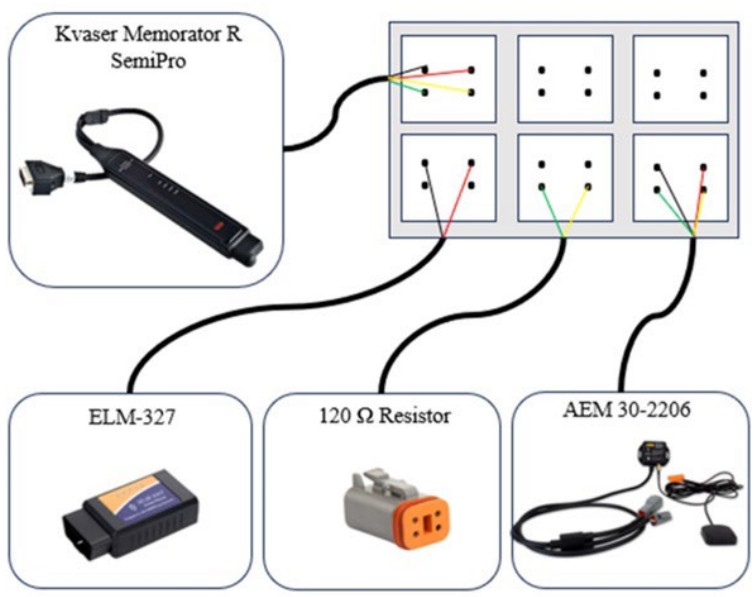

**Figure 2.** The CAN-based measurement system.

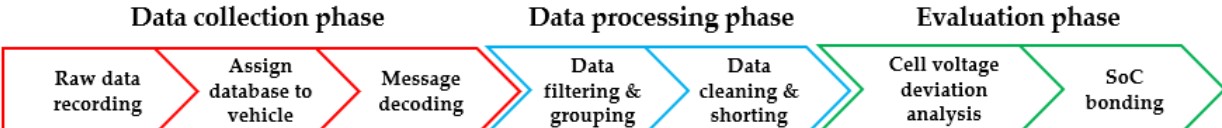

**Figure 3.** Data processing structure.

*2.2. Test Process*

The performed tests included five different vehicle battery diagnostic tests for 3 Volkswagen e-Golfs. Tests 1-2 (Test_01) and 3-4 (Test_02) were performed in pairs for better comparability, while test case 5 (Test_03) was selected in a platooning measurement with the comparison of other models. The identifiers of the cars included in the test are given in the following table (Table 1).

**Table 1.** Vehicles involved in tests.

| Serial Number | Battery Capacity | Test Days |
|:---:|:---:|:---:|
| e-Golf_1 | 35.8 kWh | Test_01, Test_02, Test_03 |
| e-Golf_2 | 35.8 kWh | Test_01 |
| e-Golf_3 | 35.8 kWh | Test_02 |

During the T1 test day, it started when the e-Golf_1 and e-Golf_2 passenger cars were at a high state of charge. The measurement system set-up and values recorded were identical for both cars, as shown in Figure 2. The measurements were run on a section of the ZalaZONE Automotive Test Track-Rural Road (Figure 4 marked in blue), where different speed limits are in force due to different environmental elements, thus simulating real road traffic. On the Rural Road element of the test track, the maximum speed limits of each section have been pre-recorded and are linked to the track regulations and the current level of safety training, as shown in Figure 4.

The length of the 2 × 1 lane element is 2500 m, while the 2 × 2 lane part is 500 m long. The following sections were used in our measurements: 90–70, 90, 60, 110, and 90 km/h. During the process, the two electric vehicles followed each other, keeping some distance between them. After completing a few laps, the cars swapped positions to prevent any wind shadow from influencing the measurement outcomes. The vehicles were driven at the maximum allowed speed, with A/C set at 20 °C and ventilation level two. The tests

were run until the vehicle's computer indicated it was fully discharged, which was 10% SoC level.

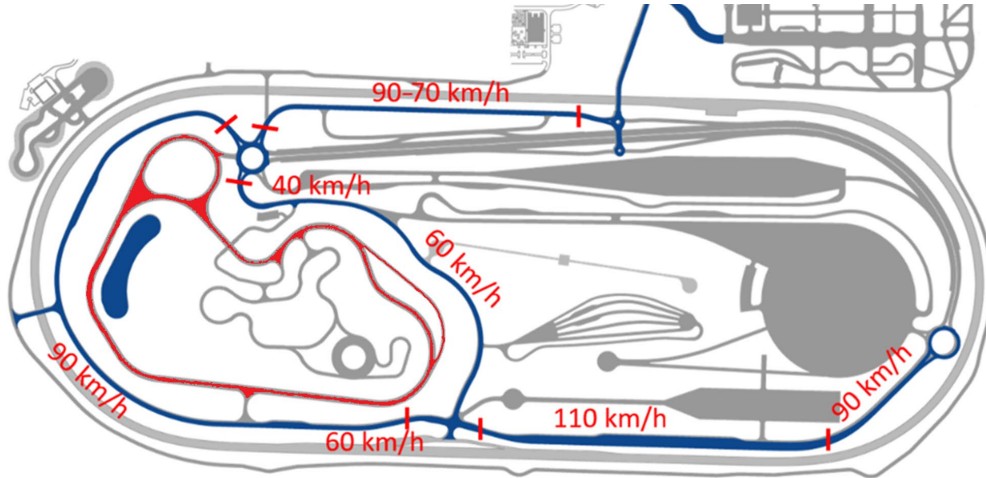

**Figure 4.** ZalaZONE Automotive Test Track-Rural Road (Blue)-Handling track (Red).

On the T2 test day, the full immersion tests were performed on the e-Golf_1 and e-Golf_3 passenger cars on the ZalaZONE-Handling track (Figure 4 marked in red), again in a paired configuration. In preparation for the measurement procedure, approximately identical SoCs based on the onboard computer were placed on the vehicles. The track element is 2000 m long and 12 m wide, with varying topography. This allows for easy reproduction of a discharge test. The component allows for various standard measurements to be carried out, with a maximum speed of 150 km/h. The two vehicles followed each other on the handling track element and changed position in 15 min cycles. No air conditioning or heating was used during the measurements, with one person in the e-Golf_1 vehicle and two in the e-Golf_3 vehicle.

In the Test_03 case, only the e-Golf_1 underwent a full immersion test, which was conducted alongside other car types as part of a platooning test. About the measurement, a position change was performed every 15 min in the sequence of the four vehicles tested, with air conditioning at 22 °C. In this case, three persons were in the vehicle. The track element used was Rural Road (Figure 4 marked in blue), which was defined similarly to Test_01, with the total discharge in this case also up to 10% SoC.

It is important to note that the method has been tested on an automotive test track. The study does not include investigating cell rebalancing strategy during charging. Furthermore, the tests do not directly interfere with the vehicle BMS equalization process or strategy.

*2.3. Calculation Method*

After processing and cleaning the data, it was necessary to examine the results of each test separately. The aim was to develop an algorithm capable of detecting and locating the weakest cell from the system voltages. For this purpose, a voltage deviation-based solution was developed. Based on the number and strength of the voltage deviations, the battery system is analyzed, and the critical cells are identified. The first step in the evaluation is determining the average voltage at each instant. This is followed by a cell-by-cell analysis, during which the deviations from the mean of 0 mV, 12 mV, 60 mV, 120 mV, and 240 mV are aggregated. Deviations with higher voltage have a higher weighting factor in the aggregation. A weighted deviation analysis method was therefore used for the implementation. Two approaches were implemented: average voltage calculation (AVC) and moving average voltage calculation (MAVC).

According to the first approach (average voltage calculation), for each measurement time point $t$, calculate the average voltage $\overline{V}_t$ across $N$ cells as follows:

$$\overline{V}_t = \frac{1}{N} \sum_{i=1}^{N} V_{it} \tag{1}$$

where $V_{it}$ is the voltage of the $i$-th cell at time $t$. Reduce the average voltage by predefined $\Delta V$ values (e.g., 0.012 V, 0.060 V, 0.120 V, 0.240 V) and evaluate the deviations from the new average:

$$\overline{V}'_t = \overline{V}_t - \Delta V \tag{2}$$

where is the reduced threshold voltage value $\overline{V}'_t$. Deviations are reevaluated based on the reduced simple average:

$$D_{it} = \begin{cases} 1, & if \left| V_{it} - \overline{V}'_t \right| > 0 \\ 0, & otherwise \end{cases} \tag{3}$$

where $D_{it}$ is a binary value and 1 for deviation from the mean. The second approach was to adjust for possible erroneous sample values. The moving average for each time point $t$ is calculated by taking the simple average of the preceding, current, and following time points:

$$MA_t = \frac{1}{3} \left( \overline{V}_{t-1} + \overline{V}_t + \overline{V}_{t+1} \right) \tag{4}$$

where $MA_t$ is the cell voltage value used, which depends on the previous, current, and following voltage values. Deviations are reevaluated based on the difference between the cell voltage and the moving average, considering the reduced simple average:

$$D_{it} = \begin{cases} 1, & if \ |V_{it} - MA_t| > \left| \overline{V}'_t - MA_t \right| \\ 0, & otherwise \end{cases} \tag{5}$$

where $D_{it}$ is a binary value and 1 for deviation from the mean. It is important to note that two methods were processed and evaluated in parallel.

The next step in the analysis was to determine the weight values. Square root transformation weighting was employed to analyze voltage deviations in batteries. Different deviations need not be weighted in equal proportion when selecting weights. More significant deviations were assigned greater weights but were not allowed to dominate excessively.

The first step involved defining the range of investigation and the magnitude of deviations. The total range under examination was between 4.2 V and 3 V, with deviations being 0 mV (0%), 12 mV (1%), 60 mV (5%), 120 mV (10%), and 240 mV (20%). The next step was to determine the transformed weights using the following formula:

$$\omega(V_{diff}) = \sqrt{V_{diff} + 1} \tag{6}$$

where $V_{diff}$ is the voltage deviation (in mV), and adding 1 (minimum value addition) ensures a weight value even for a 0 mV deviation. The calculation of weights based on this was as follows:

$$\omega(0 \text{ mV}) = \sqrt{0 + 1} = 1 \tag{7}$$

$$\omega(12 \text{ mV}) = \sqrt{12 + 1} \approx 3.606 \tag{8}$$

$$\omega(60 \text{ mV}) = \sqrt{60 + 1} \approx 7.810 \tag{9}$$

$$\omega(120 \text{ mV}) = \sqrt{120 + 1} \approx 11 \tag{10}$$

$$\omega(240 \text{ mV}) = \sqrt{240 + 1} \approx 15.524 \tag{11}$$

These weights were then re-normalized considering the following:

$$\omega_{norm}(V) = \frac{\omega(V)}{\sum \omega(V)} \tag{12}$$

where $\omega(V)$ are the individual transformed weights, and $\sum \omega(V)$ is the sum of the weights. Figure 5 displays the final weight values.

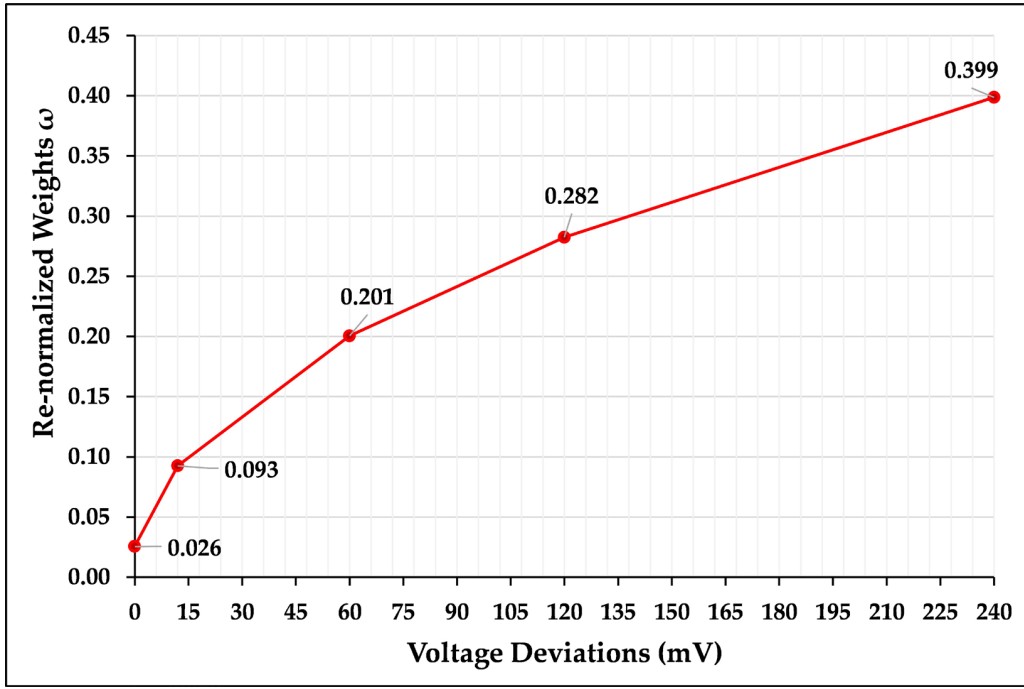

**Figure 5.** Determination of re-normalized weights based on voltage deviation.

Based on Figure 5, it is evident that the larger the voltage deviation, the greater the weight value it is assigned from the perspective of the analysis.

The deviations are separately summed for each $\Delta V$ value, applying appropriate weights ($\omega_{\Delta V}$). The weighted sum for each cell $i$ and deviation $\Delta V$ is calculated using the following formula:

$$S_{i\Delta V} = \sum_{t} \omega_{\Delta V} \cdot D_{it} \tag{13}$$

where $\omega_{\Delta V}$ is the weight associated with the given $\Delta V$ value (e.g., 0.026, 0.093, 0.201, 0.282, and 0.399).

## 3. Results and Discussion

During the tests, the battery system of Volkswagen e-Golf vehicles was tested across a large SoC range. Three vehicles were used for this purpose and tested over five complete discharge cycles. The deviations from the average voltages of each cell were analyzed, where 0 mV, 12 mV, 60 mV, 120 mV, and the range of 240 mV were determined. Another important aspect was the detection of voltage deviations as a function of charge levels, where the aim was to determine the coupling between voltage deviations and SoC. The following table shows the main benchmarks obtained in the tests performed.

In Table 2, the basic values necessary for the evaluation of the five tests can be observed. It can be seen that three different vehicles have been used for the tests (the number, e.g., e-Golf_1, can identify them). The results of Tests 1 to 5 are referenced below. Cases 1, 2, and 5 had a higher baseline charge level (approximately 95%), with average speeds of ~68 km/h in three cases and ~74 km/h in two cases. Comparing Test_01 and Test_02 in pairs, under approximately identical conditions, in a test performed simultaneously, the time spent on the course was more with the vehicle used in Test_02. This may be because

although the cars were purchased (bought) almost the same way, the vehicles used during Test_01 have more mileage (more kilometers used).

**Table 2.** Baseline values according to tests.

| Vehicle | SOC_Start [%] | Speed_AVR [km/h] | Time [s] | Energy [Wh] |
|---------|---------------|------------------|----------|-------------|
| e-Golf_1 | 94.00 | 68.48 | 8897 | 24,897 |
| e-Golf_2 | 94.80 | 68.37 | 8178 | 34,039 |
| e-Golf_1 | 86.00 | 75.33 | 5170 | 24,897 |
| e-Golf_3 | 83.60 | 73.81 | 5235 | 22,835 |
| e-Golf_1 | 95.20 | 68.16 | 8548 | 27,843 |

Taking the Test_03 and Test_04 pairs (the measurement was performed at the same time), the average speed in this case was higher than in the first pair of measurements. At the time of the test, it can be observed that although Test_4 lasted longer, the energy used was less. This is because, at the end of the test, the vehicle used for Test_4 slowed down (slower lap times), which is also shown in the average speed.

The Test_05, Test_01, and Test_03 tests were carried out with the same vehicle. The available energy was larger, but the measurement started from a higher SoC level. Compared to the other cases, the energy discharged was approximately proportional to the charge level and the average load.

Figure 6 shows the SoC and system voltage change of the different measurements.

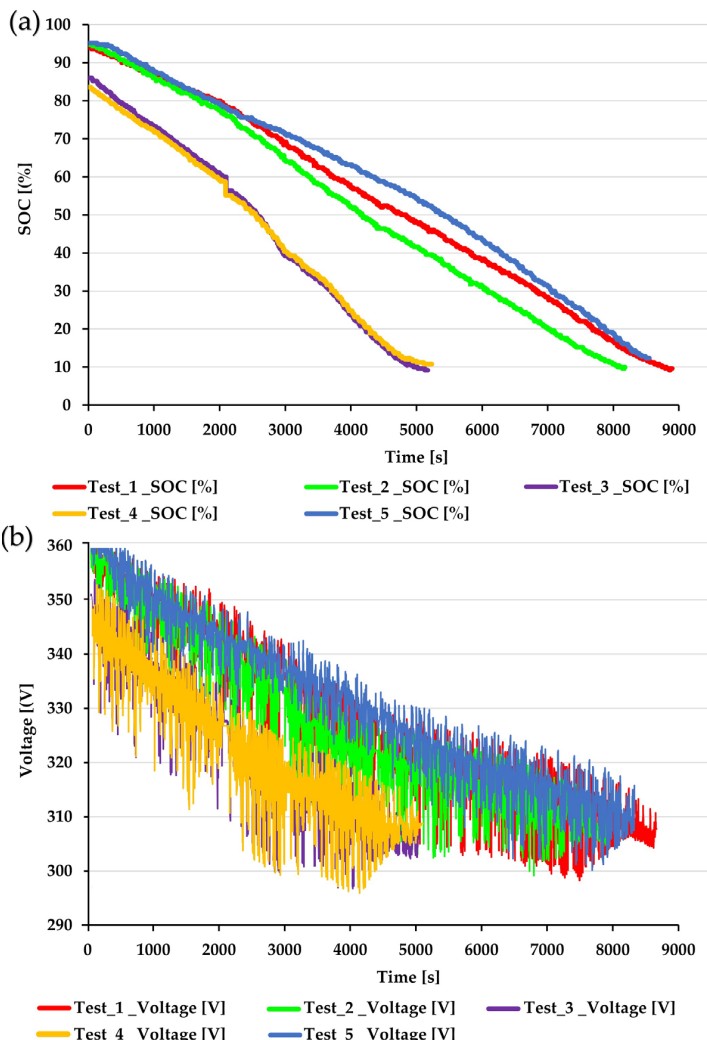

**Figure 6.** Volkswagen e-Golf during full discharge test: (**a**) SoC change; (**b**) voltage change.

Figure 6a shows the vehicles' charge level as a time function. Test_01, Test_02, and Test_05 had a higher SoC level at baseline, which resulted in a more extended test time. A dynamic change is observed at system voltage (b), caused by continuous acceleration and deceleration.

### 3.1. Detecting and Locating a Faulty Cell in the Battery System

An approach to search for weaker cells in the Volkswagen e-Golf has already been developed in a previous publication and extended to several vehicles for the full charge range [33]. The approach has been presented in Section 2.3, with a flowchart shown in Figure 3. In the procedure, all serially connected cells in the system are monitored. The weakest cells are selected based on the voltage deviations during different loads. After the measurements were taken and processed, the next step was the evaluation, where the first step was to analyze the deviation of the cells from the average voltage: 0 mV, 12 mV, 60 mV, 120 mV, and 240 mV. It is important to note that all five measurements were tested during the analysis, and all 88 cells were calculated equally. Table 3 shows the cell voltage differences.

**Table 3.** Deviations of different measurements from the average voltage.

| DIFF | Test_01 | Test_02 | Test_03 | Test_04 | Test_05 |
|------|---------|---------|---------|---------|---------|
| 0mV_AVC | 282.060 | 257.785 | 124.328 | 1.063.835 | 287.607 |
| 0mV_MAVC | 282.060 | 257.438 | 124.485 | 1.082.770 | 284.751 |
| 12mV_AVC | 135.753 | 112.844 | 56.394 | 698.301 | 108.992 |
| 12mV_MAVC | 116.865 | 112.502 | 44.281 | 125.641 | 95.146 |
| 60mV_AVC | 30.587 | 17.482 | 9.120 | 34.768 | 8708 |
| 60mV_MAVC | 22.927 | 17.140 | 4.116 | 28.082 | 6054 |
| 120mV_AVC | 7.146 | 3.960 | 2.019 | 7.733 | 703 |
| 120mV_MAVC | 4.167 | 3.618 | 638 | 5.293 | 310 |
| 240mV_AVC | 259 | 796 | 115 | 1.099 | 8 |
| 240mV_MAVC | 84 | 454 | 16 | 218 | 2 |

Table 3 shows the total number of deviations tested at the different cut-off points for all measurements. The first column shows the test criteria. Columns two, three, four, and five show the values of the different tests. The rows in the table show the number of deviations. The rows where the MAVC value is shown in the first column use the moving average method; otherwise, the deviation from the mean is applied. Comparisons between measurements one and five are most relevant, starting from similar SoC levels, and the vehicle was the same. In the 0 mV case, the number of deviations was almost the same, but at 12 mV, it was observed that there were fewer in the fifth test. For 60 mV, 120 mV, and 240 mV, much smaller values are observed in the fifth test than in the first. To determine the difference, it is possible to consider the current profile between the two measurements, which can be observed in Figure 7.

In Figure 7, the current variations in Test 1 (red) have a wider amplitude and are more frequent, while in Test 5 (blue), current variations are less frequent and have a smaller amplitude. Table 4 summarizes the values of the current changes and their standard deviation.

The current variations in Test_01 are more significant on average and variable, as reflected by the higher mean and standard deviation of the current variations. Test_05 showed more minor mean and max-average current variations. This suggests that the vehicle was subjected to more dynamic loading (more significant load fluctuations) in Test_01. The less dynamic measurement in Test_05 is also because the test was a platooning test, so no significant acceleration and deceleration were applied.

**(A) Current [A]_Test_1**

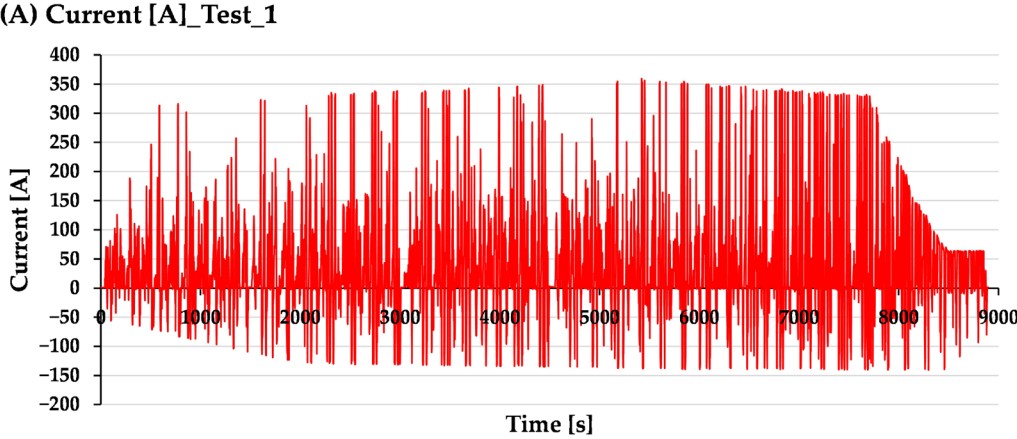

**(B) Current [A]_Test_5**

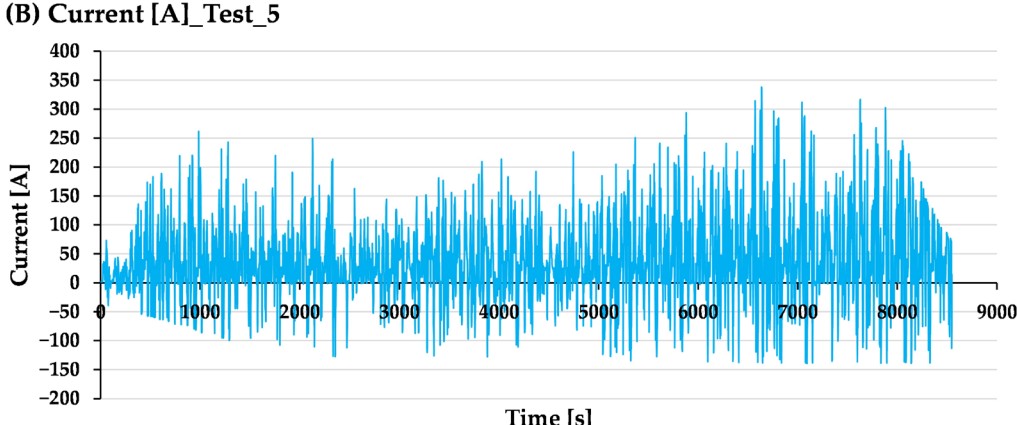

**Figure 7.** Current profiles of Test_01 and Test_05 measurements.

**Table 4.** Current variation values for e-Golf_1 during measurements one and five.

| Test Number | Average of Current Changes [A] | Standard Deviation [A] | Maximum Current Change [A] |
|---|---|---|---|
| Test_01 | 33.31 | 46.68 | 344.00 |
| Test_05 | 17.99 | 21.67 | 216.25 |

The next step in the evaluation was to observe each test cell by cell, and the results were presented in aggregate with weighting. Figure 8 shows the analysis of the first measurement.

Figure 8 shows the weighted average of the results from the first measurement. The horizontal axis shows the battery ID value, and the vertical axis indicates the number of deviations. The results are shown in red for the average value summation and green for the values obtained using moving averaging. The red line in the figure marks the critical values, defined as 10% of the sampled value per cell. This means that there were approximately 7200 data points from each cell, and if the weighted deviation was 10% off the mean voltage, it was marked as critical. A horizontal line in blue marked a deviation of 5%. The critical batteries for the first vehicle are ID 7, ID 11, ID 21, and ID 71. Figure 9 shows the results from the fourth measurement.

In the fourth test (Test_04), the third vehicle (number: e-Golf_3) was used, and it drove the most kilometers. The test method and evaluation were the same as in the previous case. The cell deviations in this case were very similar. The cells with a weighted deviation of 10% are ID 7, ID 11, ID 21, ID 23, ID 37, ID 71, and ID 87. In Table 5, the cells that were included in the faulty category in any of the five measurements are summarized.

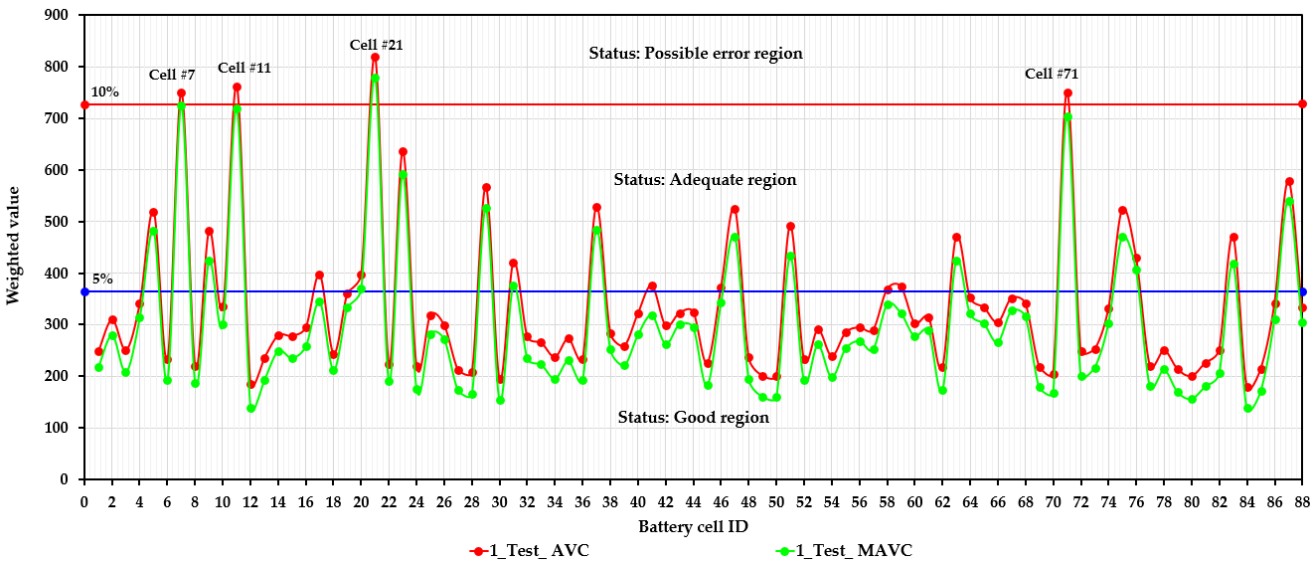

**Figure 8.** Weighted results from the first test.

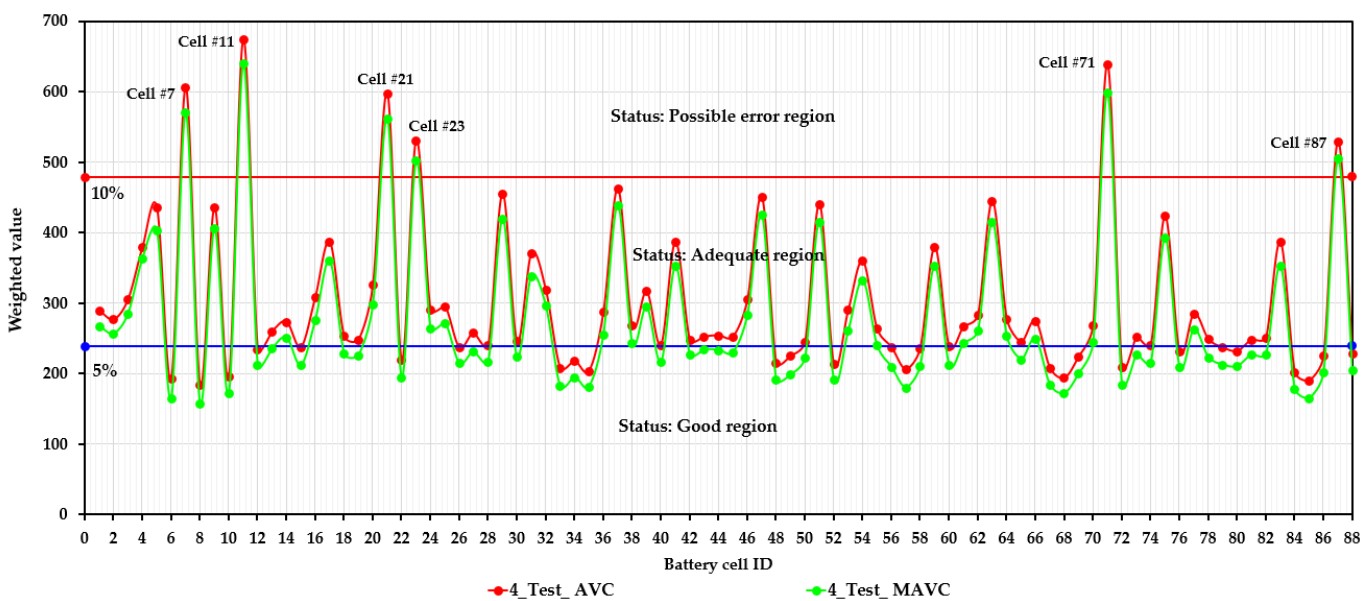

**Figure 9.** Weighted results from the fourth test.

**Table 5.** Identifying and analyzing the critical cells in the five tests.

| Test_NO | Method | Cell #7 [%] | Cell #11 [%] | Cell #21 [%] | Cell #23 [%] | Cell #71 [%] | Cell #87 [%] |
|---------|--------|-------------|--------------|--------------|--------------|--------------|--------------|
| Test_1 | AVC | **10.32** | **10.47** | **11.25** | 8.74 | **10.31** | 7.95 |
| | MAVC | 9.95 | 9.89 | **10.72** | 8.13 | 9.68 | 7.43 |
| Test_2 | AVC | 8.43 | 8.82 | 8.07 | 8.47 | 9.67 | 8.47 |
| | MAVC | 8.39 | 8.76 | 8.00 | 8.42 | 9.61 | 8.40 |
| Test_3 | AVC | **13.50** | **11.23** | **10.70** | 8.90 | **13.49** | 8.34 |
| | MAVC | **12.30** | **10.04** | 9.33 | 7.99 | **12.20** | 7.34 |
| Test_4 | AVC | **12.63** | **14.06** | **12.46** | **11.07** | **13.32** | **11.03** |
| | MAVC | **11.90** | **13.34** | **11.72** | **10.47** | **12.49** | **10.52** |
| Test_5 | AVC | 6.91 | 7.83 | 7.70 | 6.55 | 8.43 | 7.05 |
| | MAVC | 6.59 | 7.37 | 7.29 | 6.26 | 7.93 | 6.72 |

Table 5 shows cells in the possible error region for each measurement. It is important to note that data from 3 different condition vehicles are presented, so the comparison is more suitable for analyzing the characteristics of the same type. The first column of the table shows the number of measurements, and the second column shows the method used to investigate the discrepancy, AVC, and MAVC. The following six columns show the highlighted cells by ID. The deviations are expressed as a % weighted average and refer to the total discharge time. The 10% deviation is a possible error category, and the 5% is still an adequate region. This indicates that the cells have not reached the good region. However, they do not fall into the faulty region for all cases. The cells most frequently included in the incorrect category are ID 7, ID 11, ID 21, and ID 71.

### 3.2. The Relationship between Cell Voltage Deviation and the SoC

The analysis focused on the correlation between SoC levels and deviations. Total discharge tests used a 10% step size. The Figure 10 shows the cell deviations from the average voltage of the first measurement (0 mV case).

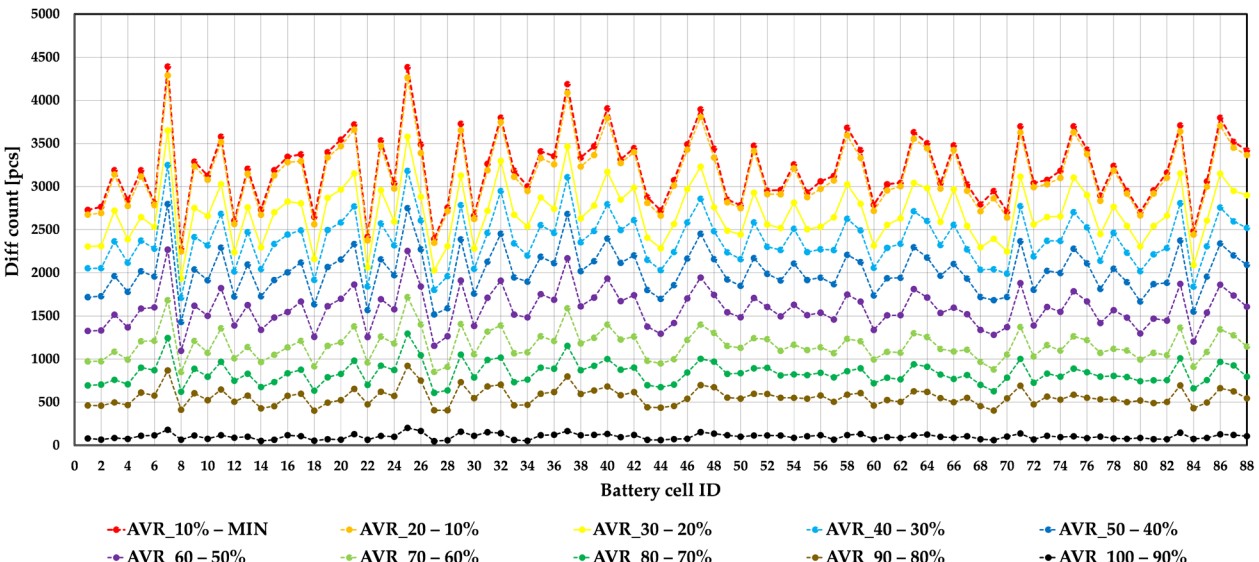

**Figure 10.** Deviation of 0 mV in the first measurement as a function of SoC.

The horizontal axis of Figure 10 shows the cell identifiers (ID), and the vertical axis shows the deviation number. The first case under investigation deviates from the mean value of 0 mV. The cases in the figure refer to different SoC levels. Under_AVR_MIN indicates the number of deviations observed up to the fully discharged state. Under_AVR_10% shows the number of deviations up to the last 10%, and Under_AVR_20% is the sum of cases up to 20%. The values of each deviation are added continuously. Thus, the solid lines in Figure 11 present the total deviation up to the specified SoC level. In the Figure 11 the deviations at 60 mV are observed.

The diagram's structure in Figure 11 is similar to the previous one, and the different SoC levels are separated. In this case, much smaller values are observed. There are fewer cases when the average voltage value is reduced to 60 mV. In Figure 12, the outlier was investigated, where a deviation from 240 mV was monitored.

Figure 12 shows that 240 mV already deviates from a few cells, and not in many cases. However, it can be seen that this is not occurring at a particular SoC value level. The additional measurements are shown in a combined form. The analysis covers all five tests, AVC and MAVC cases, and all ten SOC levels. Figure 13 shows the heat map of the % deviation from different voltage levels.

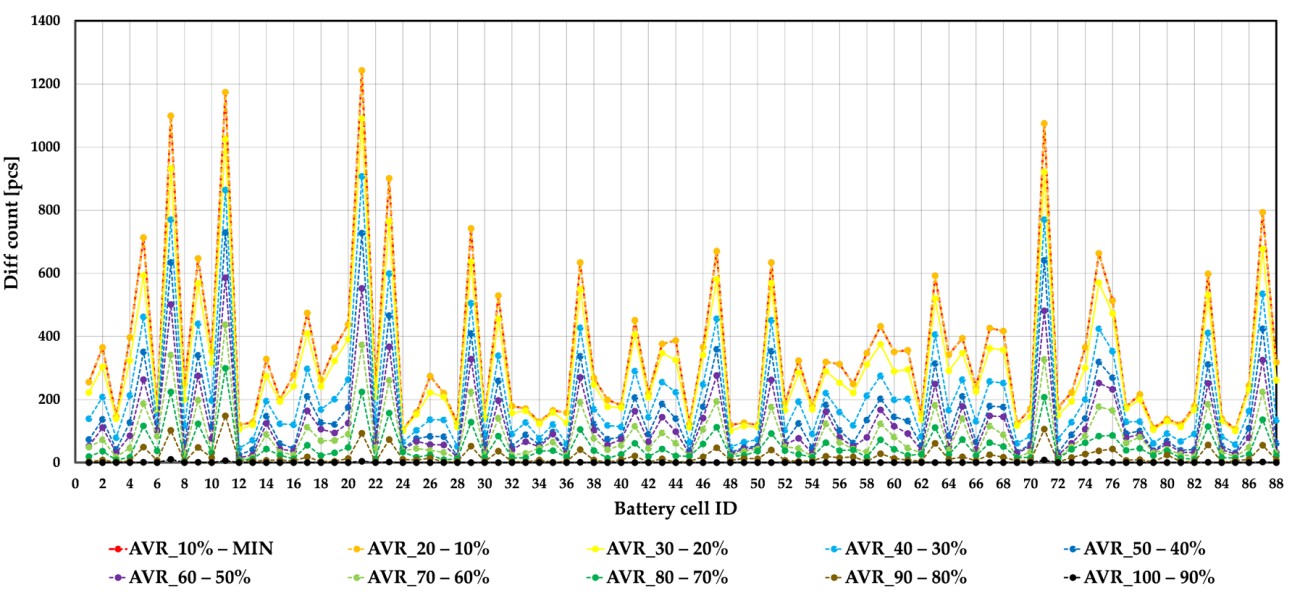

**Figure 11.** Deviation of 60 mV in the first measurement as a function of SoC.

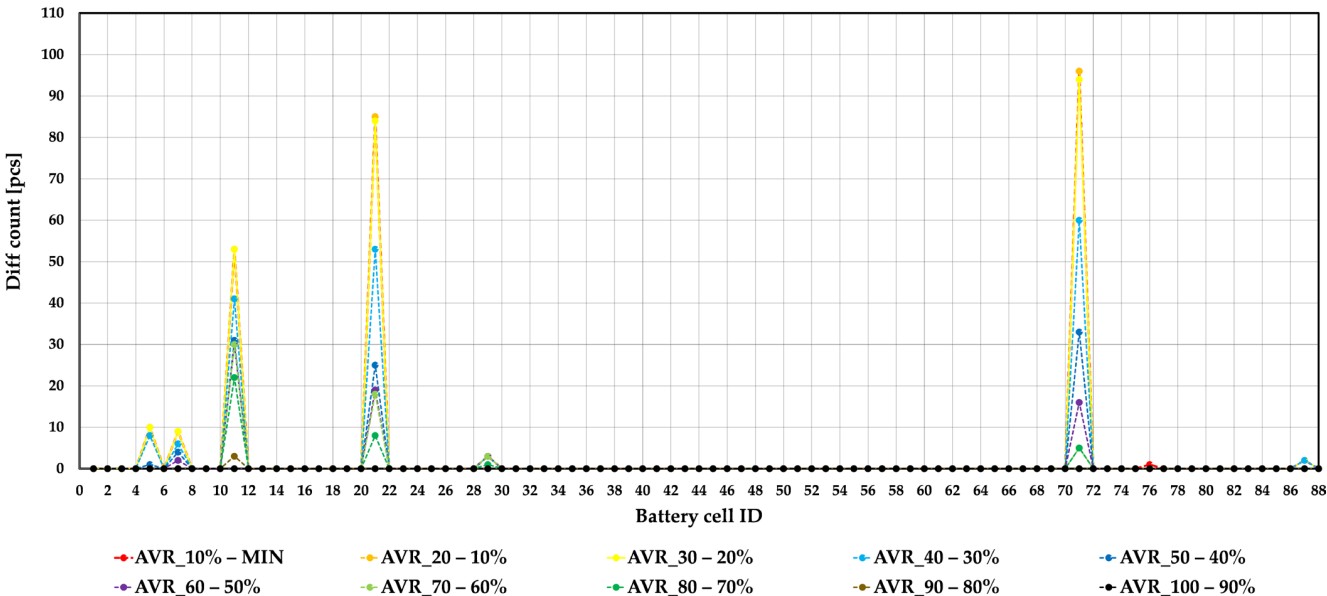

**Figure 12.** Deviation of 240 mV in the first measurement as a function of SoC.

Figure 13 summarizes all the measurement results, with different parts divided according to the degree of deviation. Therefore, in Figure 13, part (A) shows the 0 mV deviations of all measurements, part (B) shows the 12 mV, and so on. The first column of the heat map (table) structure in the figure shows the definition of the measurement and evaluation method. In the table row structure, the same deviation calculations have been associated with the corresponding deviation calculations of the different measurements, both AVC and MAVC. In columns 2 to 10 of each heat map, the % SoC for the different measurements is displayed. The data in the rows of the table are presented by total deviation percentage (each row totals 100%). For example, the second row of the heat map shows the 'Test_1_AVC' evaluation, which shows the distribution of charge levels for the analysis at one voltage deviation level (e.g., (B) 12 mV). Higher deviations are shown in red, while lower deviations are shown in green on the heat map. Table 6 summarizes the % of all SoCs.

**(A) Results 0 mV deviation**

| Test_No | MIN | 10% | 20% | 30% | 40% | 50% | 60% | 70% | 80% | 90% |
|---|---|---|---|---|---|---|---|---|---|---|
| Test_1_AVC | 2.01 | 13.82 | 9.79 | 11.52 | 13.40 | 13.31 | 10.03 | 8.63 | 14.36 | 3.13 |
| Test_1_MAVC | 1.90 | 13.58 | 10.15 | 11.83 | 13.40 | 13.43 | 9.95 | 8.76 | 14.01 | 2.98 |
| Test_2_AVC | 1.78 | 12.71 | 10.77 | 11.54 | 12.14 | 9.39 | 8.60 | 10.48 | 15.34 | 7.27 |
| Test_2_MAVC | 1.67 | 12.72 | 10.80 | 11.56 | 12.15 | 9.40 | 8.60 | 10.51 | 15.35 | 7.26 |
| Test_3_AVC | 4.77 | 14.51 | 9.75 | 12.89 | 6.59 | 9.88 | 16.17 | 16.43 | 8.98 | 0.03 |
| Test_3_MAVC | 4.65 | 14.33 | 9.66 | 12.91 | 6.45 | 9.78 | 16.67 | 16.66 | 8.85 | 0.03 |
| Test_4_AVC | 18.72 | 18.72 | 18.72 | 12.93 | 10.55 | 8.79 | 6.58 | 3.95 | 1.03 | 0.00 |
| Test_4_MAVC | 18.69 | 18.69 | 18.69 | 13.02 | 10.57 | 8.80 | 6.58 | 3.94 | 1.03 | 0.00 |
| Test_5_AVC | 4.03 | 3.08 | 8.85 | 9.21 | 10.96 | 12.83 | 13.98 | 14.97 | 13.44 | 8.66 |
| Test_5_MAVC | 4.07 | 3.16 | 8.94 | 9.20 | 11.08 | 12.86 | 13.91 | 14.97 | 13.23 | 8.57 |

**(B) Results 12 mV deviation**

| Test_No | MIN | 10% | 20% | 30% | 40% | 50% | 60% | 70% | 80% | 90% |
|---|---|---|---|---|---|---|---|---|---|---|
| Test_1_AVC | 1.37 | 15.90 | 13.94 | 13.73 | 13.16 | 13.61 | 9.80 | 8.88 | 8.93 | 0.68 |
| Test_1_MAVC | 1.80 | 16.28 | 12.37 | 12.01 | 11.72 | 9.98 | 9.10 | 11.22 | 12.44 | 3.09 |
| Test_2_AVC | 1.82 | 12.47 | 13.17 | 12.81 | 13.42 | 11.31 | 9.74 | 10.55 | 12.16 | 2.54 |
| Test_2_MAVC | 1.54 | 12.50 | 13.23 | 12.85 | 13.47 | 11.33 | 9.77 | 10.58 | 12.22 | 2.51 |
| Test_3_AVC | 2.40 | 12.88 | 12.19 | 11.84 | 8.19 | 6.72 | 19.82 | 17.78 | 8.18 | 0.00 |
| Test_3_MAVC | 2.17 | 12.99 | 11.41 | 11.15 | 8.00 | 6.45 | 21.93 | 17.85 | 8.04 | 0.00 |
| Test_4_AVC | 19.04 | 19.04 | 19.04 | 13.25 | 10.65 | 8.41 | 6.33 | 3.50 | 0.74 | 0.00 |
| Test_4_MAVC | 0.00 | 0.00 | 30.19 | 13.93 | 11.93 | 11.34 | 15.35 | 13.92 | 3.34 | 0.00 |
| Test_5_AVC | 5.91 | 5.21 | 12.50 | 13.61 | 13.99 | 12.94 | 13.71 | 9.91 | 9.00 | 3.22 |
| Test_5_MAVC | 6.14 | 5.72 | 13.38 | 13.77 | 14.30 | 12.91 | 13.81 | 9.15 | 8.00 | 2.81 |

**(C) Results 60 mV deviation**

| Test_No | MIN | 10% | 20% | 30% | 40% | 50% | 60% | 70% | 80% | 90% |
|---|---|---|---|---|---|---|---|---|---|---|
| Test_1_AVC | 0.00 | 12.24 | 25.13 | 17.47 | 9.92 | 10.32 | 10.34 | 9.06 | 5.39 | 0.13 |
| Test_1_MAVC | 0.01 | 12.56 | 26.95 | 17.70 | 9.66 | 10.04 | 11.36 | 8.36 | 3.33 | 0.03 |
| Test_2_AVC | 3.83 | 7.60 | 13.61 | 14.86 | 13.75 | 13.35 | 13.12 | 11.13 | 7.60 | 1.15 |
| Test_2_MAVC | 1.91 | 7.75 | 13.93 | 15.16 | 14.04 | 13.58 | 13.37 | 11.33 | 7.81 | 1.11 |
| Test_3_AVC | 0.04 | 7.59 | 17.02 | 12.68 | 13.91 | 6.18 | 19.34 | 14.52 | 8.72 | 0.00 |
| Test_3_MAVC | 0.02 | 7.56 | 17.15 | 13.34 | 18.44 | 8.62 | 15.09 | 11.59 | 8.19 | 0.00 |
| Test_4_AVC | 0.00 | 0.00 | 33.32 | 12.52 | 20.97 | 8.88 | 13.27 | 9.27 | 1.77 | 0.00 |
| Test_4_MAVC | 0.00 | 0.00 | 32.67 | 13.40 | 22.19 | 8.64 | 13.72 | 8.09 | 1.29 | 0.00 |
| Test_5_AVC | 3.93 | 12.02 | 21.16 | 21.76 | 16.55 | 8.81 | 6.96 | 3.57 | 4.21 | 1.02 |
| Test_5_MAVC | 2.87 | 13.31 | 21.72 | 23.01 | 16.86 | 8.51 | 6.71 | 2.79 | 3.75 | 0.46 |

**(D) Results 120 mV deviation**

| Test_No | MIN | 10% | 20% | 30% | 40% | 50% | 60% | 70% | 80% | 90% |
|---|---|---|---|---|---|---|---|---|---|---|
| Test_1_AVC | 0.01 | 8.02 | 31.60 | 20.72 | 7.95 | 8.76 | 11.32 | 9.01 | 2.60 | 0.00 |
| Test_1_MAVC | 0.07 | 8.28 | 33.26 | 20.23 | 8.71 | 8.52 | 12.02 | 7.63 | 1.27 | 0.00 |
| Test_2_AVC | 16.92 | 3.43 | 9.49 | 14.49 | 11.14 | 12.73 | 14.04 | 10.81 | 6.16 | 0.78 |
| Test_2_MAVC | 9.07 | 3.76 | 10.39 | 15.92 | 12.35 | 13.79 | 15.28 | 11.83 | 6.80 | 0.80 |
| Test_3_AVC | 0.00 | 4.66 | 19.81 | 14.66 | 19.71 | 7.73 | 12.18 | 14.96 | 6.29 | 0.00 |
| Test_3_MAVC | 0.00 | 6.27 | 21.32 | 13.95 | 30.25 | 8.93 | 5.17 | 8.31 | 5.80 | 0.00 |
| Test_4_AVC | 0.00 | 0.00 | 37.14 | 11.26 | 26.76 | 6.76 | 11.82 | 5.60 | 0.66 | 0.00 |
| Test_4_MAVC | 0.00 | 0.00 | 31.95 | 11.58 | 30.36 | 7.25 | 13.00 | 5.50 | 0.36 | 0.00 |
| Test_5_AVC | 0.00 | 11.95 | 26.74 | 38.26 | 14.79 | 1.71 | 2.28 | 2.56 | 1.71 | 0.00 |
| Test_5_MAVC | 0.32 | 10.65 | 27.42 | 45.81 | 11.61 | 0.32 | 2.58 | 1.29 | 0.00 | 0.00 |

**(E) Results 240 mV deviation**

| Test_No | MIN | 10% | 20% | 30% | 40% | 50% | 60% | 70% | 80% | 90% |
|---|---|---|---|---|---|---|---|---|---|---|
| Test_1_AVC | 0.39 | 1.16 | 31.66 | 29.34 | 10.42 | 5.41 | 7.72 | 12.74 | 1.16 | 0.00 |
| Test_1_MAVC | 3.57 | 0.00 | 21.43 | 32.14 | 15.48 | 5.95 | 7.14 | 13.10 | 1.19 | 0.00 |
| Test_2_AVC | 84.17 | 0.00 | 0.63 | 3.14 | 2.76 | 2.14 | 4.52 | 2.39 | 0.25 | 0.00 |
| Test_2_MAVC | 72.25 | 0.00 | 1.10 | 5.51 | 4.85 | 3.74 | 7.93 | 4.19 | 0.44 | 0.00 |
| Test_3_AVC | 0.00 | 0.00 | 14.78 | 21.74 | 48.70 | 7.83 | 4.35 | 2.61 | 0.00 | 0.00 |
| Test_3_MAVC | 0.00 | 0.00 | 0.00 | 0.00 | 87.50 | 12.50 | 0.00 | 0.00 | 0.00 | 0.00 |
| Test_4_AVC | 0.00 | 0.00 | 72.34 | 4.55 | 16.47 | 1.91 | 3.55 | 1.18 | 0.00 | 0.00 |
| Test_4_MAVC | 0.00 | 0.00 | 16.51 | 10.09 | 53.67 | 7.34 | 8.72 | 3.67 | 0.00 | 0.00 |
| Test_5_AVC | 0.00 | 0.00 | 0.00 | 100.00 | 0.00 | 0.00 | 0.00 | 0.00 | 0.00 | 0.00 |
| Test_5_MAVC | 50.00 | 0.00 | 0.00 | 50.00 | 0.00 | 0.00 | 0.00 | 0.00 | 0.00 | 0.00 |

**Figure 13.** Heat map of % deviations from different voltage levels.

**Table 6.** The different deviations are weighted according to the SoC.

| Test_No | MIN | 10% | 20% | 30% | 40% | 50% | 60% | 70% | 80% | 90% |
|---|---|---|---|---|---|---|---|---|---|---|
| 0 mV | 6.23 | 12.53 | 11.61 | 11.66 | 10.73 | 10.85 | 11.11 | 10.93 | 10.56 | 3.79 |
| 12 mV | 4.22 | 11.30 | 15.14 | 12.90 | 11.88 | 10.50 | 12.94 | 11.33 | 8.30 | 1.49 |
| 60 mV | 1.26 | 8.06 | 22.27 | 16.19 | 15.63 | 9.69 | 12.33 | 8.97 | 5.21 | 0.39 |
| 120 mV | 2.64 | 5.70 | 24.91 | 20.69 | 17.36 | 7.65 | 9.97 | 7.75 | 3.17 | 0.16 |
| 240 mV | 21.04 | 0.12 | 15.85 | 25.65 | 23.98 | 4.68 | 4.39 | 3.99 | 0.30 | 0.00 |
| SUM (Weighted) | 9.95 | 4.65 | 19.53 | 20.83 | 18.99 | 7.23 | 8.53 | 6.92 | 3.11 | 0.36 |

Table 6 shows a summary of the results. In the first column, the size of the deviations is shown. The following ten columns summarize the % variance across the different SoC levels. It is important to note that the values in the single rows have been aggregated and averaged. For example, the values in column MIN in Figure 13A have been averaged. The values in the last row of the table have been weighted according to the weights shown in Figure 5 and Formula (13) (Section 2.3 Calculation method). The results show that it is not at the lowest SoC level that most of the deviation occurs (although this would have been the expected value). The cells with the highest probability of deviation and weaker cells are found in the 40–20% range. So, this range is where dynamic measurements should be taken for short, test-in self-diagnostics. The applicability of the method to autonomous vehicles can be significantly affected by the potential limitations of dynamic measurements. Further, continuous human control is recommended during this type of measurement.

## 4. Conclusions

This article describes the diagnosis of three different electric vehicle battery systems. A full immersion test was carried out on all three vehicles to provide an assessment of their condition. This means that the vehicles started from a high-charged state and were driven around the test track until the batteries were discharged. A fault detection and localization algorithm was applied over the entire charge level range for the evaluation. The analyses examined the deviations of cell voltages from the average, covering 0 mV, 12 mV, 60 mV, 120 mV, and 240 mV. The results were analyzed in detail, and the algorithm was used to identify cells that appeared to be weaker. It is important to note that three measurements were carried out with each vehicle, so no accurate conclusions can be reached. However, these data provide guidance and may be helpful for further diagnostic steps.

In the current measurement, the red line, i.e., the potentially faulty range, was mostly found to be ID 7, ID 11, ID 21, and ID 71. This does not necessarily indicate a fault; it may also indicate a weaker or faster aging (more sensitive) battery. To confirm this, further measurements with more vehicles are required.

An important observation for the battery self-diagnostics of electric and hybrid vehicles (even autonomous) is that the variance is likely related to the degree of load (higher number of accelerations). Thus, more dynamic tests with more effective search are recommended (as far as possible during safe autonomous operation). Furthermore, analyses in the SoC range have shown that performing tests only at the lowest charge level is not necessarily worthwhile. From a diagnostic point of view, the 40–20% range may be ideal. Therefore, to perform diagnostics, it is recommended to perform a dynamic test of at least 15 min, starting at 40–30% SoC and ending at around 20%.

Adaptation may not be exactly the same for different vehicle types, as battery systems may differ. The aim is to develop a simple but efficient approach to perform battery diagnostics with fewer computational resources. This method may hold potential for energy management and monitoring the battery condition of vehicles but is currently at the stage of a research project rather than for industrial-scale maintenance procedures. Safety critical systems such as BMS control have remained unchanged; the research has focused on data collection and its analysis. Further work related to battery condition (SoH) and deviations will require analyzing more data and battery systems in different states.

**Author Contributions:** Conceptualization, S.K.S.; methodology, S.K.S. and G.S.; software, S.K.S., G.S. and P.Ő.; validation, S.K.S.; formal analysis, S.K.S.; investigation, S.K.S. and I.L.; resources, I.L.; data curation, S.K.S., G.S. and P.Ő.; writing—original draft preparation, S.K.S. and G.S.; writing—review and editing, S.K.S.; visualization, S.K.S. and G.S.; supervision, S.K.S. and I.L.; project administration, S.K.S. and I.L.; funding acquisition, S.K.S. and I.L. All authors have read and agreed to the published version of the manuscript.

**Funding:** The research was supported by the European Union within the framework of the National Laboratory for Autonomous Systems (RRF-2.3.1-21-2022-00002).

**Data Availability Statement:** Data are contained within the article.

**Conflicts of Interest:** The authors declare no conflicts of interest.

## Abbreviations

| | |
|---|---|
| SoC | State of Charges |
| UAV | Unmanned Aerial Vehicle |
| AI | Artificial Intelligence |
| ADAS | Advanced Driver Assistance Systems |
| ISS | Integrated Self-diagnostic Systems |
| IoT | Internet of Things |
| RUL | Remaining Useful Life |
| BMS | Battery Management System |
| OBD | Onboard Diagnostics |

| | |
|---|---|
| NMC | Lithium–Nickel–Manganese–Cobalt |
| AVC | Average Voltage Calculation |
| MAVC | Moving Average Voltage Calculation |
| Test_ AVC | The AVC procedure was used to evaluate the test. |
| Test_ MAVC | The MAVC procedure was used to evaluate the test |
| AVR_X% (ex. AVR_80–70%) | Battery SoC level between 80% and 70% |
| cell ID | Number (location) of cells in the battery system |
| deviations 0 mV (0%) | Deviation from the average voltage. |
| deviations 12 mV (1%) | The deviation from the average voltage is 12 mV, which represents 1% of the voltage range tested. |

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
