# Peer review of "Self-Diagnostic Opportunities for Battery Systems in Electric and Hybrid Vehicles"

_machines, doi:10.3390/machines12050324_

Round 1
Reviewer 1 Report
Comments and Suggestions for Authors
The manuscript deals with an interesting topic on battery fault diagnosis in EVs. The comments are given below.
(1) The topic is important. However, it could be confusing to me that what is the real connections between your work in this paper and autonomous vehicles? It is well-known that autonomous vehicles could have better capability to implement such online diagnostic systems, as they are normally equipped with better computational devices and more sensors. In your work, I did not figure out the significant difference between implementing your methods to normal EVs and autonomous vehicles. Is it just because your methods hold the potential to be deployed for autonomous vehicles? You may need to explain this point.
(2) The whole paper is basically based on simple experiments. It may lack of scientific explorations. For instance, what causes the abnormal situation? Shall we use battery models to explain these phenomenon? What are the core innovative points?
(3) Did you change the balancing strategy in BMS of the vehicle? Authors may need to illustrate the method of voltage balancing for cells in the battery system. The diagnostic methods proposed here are basically based on counting the situations when cell voltages evidently deviated from the average. If the balancing strategy in BMS works, it could be a confounder to the proposed algorithm. Authors may need to explain this.
(4) The thresholds should not be chosen based on randomly setting a number. Authors may need to provide more evidence for the reason. It is doubtful that this method could be usefully deployed for different types of vehicles, given thresholds are set in this way.
(5) The average is calculated based on the voltages of all the cells in the battery pack. This means the voltages that severely deviate from the average are used for the determination of reference. This question has been raised in some literature and solved through multiple steps of screening the voltages. Authors may need to consider what will happen to your model if some cell voltages significantly deviate from the average.
(6) “Under_AVR_10% gives information on the number of deviations up to the last 10%, and Under_AVR_20% is the sum of the number of cases up to 20%, respectively.” In Figures 10–12, is it correct that the “Under_AVR_20%” is counting the abnormal situations between SOC in a range of 0% to 20%? If it is true, why Under_AVR_90% not holds the highest values? I guess that could not be the case, so I suggest better clarifying it.
(7) I strongly recommend authors should standardize their words for abbreviations in main narratives, experiment descriptions, and figures. It could be helpful to use a table or anything similar to summarize your abbreviations in your paper. It’s a bit challenging to search every abbreviation. For instance, in Figure 9, the use of “4_Test_UO” and “4_Test_AB” are not intuitive and takes readers longer time to understand. Moreover, copying abbreviations in your code as names in your scientific article is easy for you but may cause difficulty of quick understanding for readers.
(8) The authors may need to specifically tell readers which figure you are indicating in your main text. For instance, where is Figure 8 in your main text?
Overall, the topic of this manuscript is meaningful, but some important aspects need to be improved.
Comments on the Quality of English LanguageNeed to be improved.
Author Response
Thank you very much for your comments, please find the attached file.

Reviewer 2 Report
Comments and Suggestions for Authors
This paper is interesting by using e-Golf data for weakest cell detection in real tests. This paper can be accepted after minor revisions.
1. SOH and RUL are hot topics and are influenced by the inconsistencies. More accurate and reliable predictions are helpful for predictive maintenance. Please see 10.1016/j.ress.2023.109603 and 10.1109/TII.2022.3206776 for more discussions.
2. The figures need to be improved to be clearer.
3. The introduction requires improvement to be more concise to discuss the most relevant works.
4. The main research gaps and the corresponding contributions should be clearly presented.
Comments on the Quality of English LanguageMinor issues need to be addressed.
Author Response

(The authors gave the same response as above.)

Reviewer 3 Report
Comments and Suggestions for Authors
The authors researched self-inspection diagnostic technology for electric vehicle battery systems and proposed a method for locating faulty batteries based on battery voltage deviation analysis. This study contains some interesting findings and are valuable for the understanding of the fault diagnosis of autonomous vehicle. However, lack of comparative experiments with other methods is the major flaw of the study. Therefore, MAJOR/MINOR revision must be done before this manuscript could be accepted for publication in the Machines.
Major comments:
Q1. The work only tested three vehicles of the same model, with a small size of sample, and the reliability of the statistical conclusions needs to be improved. Suggest adding more vehicle models and battery pack tests.
Q2. In the introduction section, the authors need to provide detailed information on innovation points and contributions.
Q3. The description of the experimental process is not clear enough, and it is necessary to explain what is the full immersion testing.
Q4. The results are convincing but lack comparison with other methods.
Q5. The work does not provide the process of self-inspection methods in practical operation, and there is no in-depth discussion on how to balance safety and diagnostic efficiency.
Minor comments:
Q1. The reference format needs to be unified.
Q2. The numbering of the formula should be complete.
Q3. The meaning of the heat map is unclear; it is recommended to explain it.
Comments on the Quality of English LanguageIt is suggested to polish it.
Author Response

(The authors gave the same response as above.)

Round 2
Reviewer 1 Report
Comments and Suggestions for Authors
The manuscript has been improved. The only point concerns me is this work has not been directly deployed for autonomous vehicles but the title and abstract are explicitly saying autonomous vehicles.
Comments on the Quality of English LanguageFine
Author Response
Dear Editors, Dear Reviewers,
We are very grateful to you for reviewing our article on machines (MDPI) for the second time. Based on your feedback, we have improved our manuscript and hope it will now be recommended for publication. Finally, we appreciate your time and effort in providing these helpful reviews.
Response to Reviewer #1's Comments
The manuscript has been improved. The only point concerns me is this work has not been directly deployed for autonomous vehicles but the title and abstract are explicitly saying autonomous vehicles. In the rest of the article, we have also tried to highlight that the method can be used for autonomous vehicles, but it has not been used in concrete terms.
- Thank you for your comment; we accept your concern. We have changed the title and the abstract accordingly. In the rest of the article, we have also highlighted that the method can be used for autonomous vehicles, but it has not been used in concrete terms.
Reviewer 3 Report
Comments and Suggestions for Authors
The authors have responsed all of my comments.
Comments on the Quality of English LanguageIt should be polished.
Author Response
Dear Editors, Dear Reviewers,
We are very grateful to you for reviewing our article on machines (MDPI) for the second time. Based on your feedback, we have improved our manuscript and hope it will now be recommended for publication. Finally, we appreciate your time and effort in providing these helpful reviews.
Response to Reviewer #3's Comments
The authors have responsed all of my comments. Comments on the Quality of English Language It should be polished.
- Thank you for your comments and the feedback that we have made the changes you requested. We have checked the entire article and made several corrections to the language and editorial style to make it easier to understand.